# Current Treatments and New Possible Complementary Therapies for Epithelial Ovarian Cancer

**DOI:** 10.3390/biomedicines10010077

**Published:** 2021-12-31

**Authors:** Maritza P. Garrido, Allison N. Fredes, Lorena Lobos-González, Manuel Valenzuela-Valderrama, Daniela B. Vera, Carmen Romero

**Affiliations:** 1Laboratorio de Endocrinología y Biología de la Reproducción, Hospital Clínico Universidad de Chile, Santiago 8380456, Chile; allison.fredes@ug.uchile.cl (A.N.F.); daniela.vera@gmail.com (D.B.V.); 2Departamento de Obstetricia y Ginecología, Facultad de Medicina, Universidad de Chile, Santiago 8380453, Chile; 3Centro de Medicina Regenerativa, Facultad de Medicina, Clínica Alemana-Universidad del Desarrollo, Santiago 7710162, Chile; llobos@udd.cl; 4Laboratorio de Microbiología Celular, Instituto de Investigación y Postgrado, Facultad de Ciencias de la Salud, Universidad Central de Chile, Santiago 8320000, Chile; manuel.valenzuela@ucentral.cl

**Keywords:** epithelial ovarian cancer, drug repurposing, non-coding RNAs, nanocarriers, anti-angiogenic therapy

## Abstract

Epithelial ovarian cancer (EOC) is one of the deadliest gynaecological malignancies. The late diagnosis is frequent due to the absence of specific symptomatology and the molecular complexity of the disease, which includes a high angiogenesis potential. The first-line treatment is based on optimal debulking surgery following chemotherapy with platinum/gemcitabine and taxane compounds. During the last years, anti-angiogenic therapy and poly adenosine diphosphate-ribose polymerases (PARP)-inhibitors were introduced in therapeutic schemes. Several studies have shown that these drugs increase the progression-free survival and overall survival of patients with ovarian cancer, but the identification of patients who have the greatest benefits is still under investigation. In the present review, we discuss about the molecular characteristics of the disease, the recent evidence of approved treatments and the new possible complementary approaches, focusing on drug repurposing, non-coding RNAs, and nanomedicine as a new method for drug delivery.

## 1. Introduction

Anti-tumoral therapies are constantly evolving, mainly because of growing evidence about the physiopathological mechanisms involved in tumoral progression and drug resistance [1]. The approved treatments for ovarian cancer, one of the deadliest gynaecological malignancies, are limited and have had few changes in the last decades compared to the observed progress in therapies against other cancers [2]. The introduction of anti-angiogenic therapy [3] and poly adenosine diphosphate-ribose polymerases (PARP)-inhibitors [4] have been the most recent changes on therapeutic schemes, increasing the progression-free survival of EOC patients but with some important drawbacks. At present, there are many proposals for complementary therapies to the existent, including other anti-angiogenic compounds, immune checkpoint inhibitors, tropomyosin receptor kinases (TRK)-inhibitors, biological compounds as non-coding RNAs, and drug repositioning, which will be reviewed in the following sections. In addition, advances in drug delivery encourage the use of previously discarded drugs due to their toxicity or hydrophobic properties. The current work aims to provide an overview of current and potential new therapeutic options that are being tested in ovarian cancer, considering both preclinical and clinical evidence.

## 2. Epithelial Ovarian Cancer

Ovarian cancer is the most lethal gynaecological cancer worldwide. According to the American Cancer Society, in the United States, more than 21,400 women will receive a new diagnosis of ovarian cancer, and more than 13,700 women will die from ovarian cancer each year [5]. Unfortunately, there is no reliable test to screen for ovarian cancer, and symptoms are often confused with other diseases, which delays the diagnoses and treatments and results in poor survival rates [6]. 

Ovarian cancer staging is used to predict clinical behaviour and to select the appropriate therapeutic approach for patients. There are two main criteria: (1) the tumour-node-metastasis (TNM) system, based on tumour size, local growth (T), the extent of lymph node metastases (N) and occurrence of distant metastases (M) [7], and (2) the International Federation of Gynecology and Obstetrics (FIGO) classification. This system considers the fallopian tube and peritoneal origins of ovarian tumours collectively. The classification is based on the location, compromise of lymph nodes, peritoneal dissemination, ascites and metastasis to extra-abdominal organs, ranging from stage I through stage IV [8,9].

Around 85 to 90% of ovarian cancers have an epithelial origin (EOC) [10,11]. Half of them correspond to serous carcinomas, 10% to endometrioid subtype, and about 6% to clear cell and mucinous carcinoma [10]. EOC has heterogeneous nature and can be classified depending on its morphologic and molecular features. Based on that, the dualistic model confirms two major histologic types of EOC, type I and type II [12,13]. Type I tumours develop in a stepwise progression from well-established precursor lesions, such as borderline tumours and endometriosis lesions that in turn originate from cystadenomas and adenofibromas (low-grade serous carcinomas, low-grade endometrioid, clear cell, malignant Brenner tumour and mucinous carcinomas) [12,14]. These neoplasms are present as large masses, confined to one ovary with better prognosis but, importantly, seem not to respond well to adjutant chemotherapy, being the optimal debulking surgery the best option [15,16,17]. 

On the other hand, type II carcinomas evolve rapidly, are highly aggressive with rapid growth, and tend to spread sooner [12]. These tumours are relatively sensitive to platinum and taxane-based chemotherapy [18]. Some examples of type II carcinomas are high-grade serous (HGS) EOC, high-grade endometrioid carcinoma, carcinosarcomas and undifferentiated carcinomas [12,14]. 

It is commonly proposed that serous tumours, the more frequent histological type of EOC, derive from two origins, cortical inclusion cysts from ovarian surface epithelium or malignant precursors from fimbrial epithelium [19,20,21]. Fimbrial cells could be implanted on the disrupted ovarian surface forming inclusion cysts, and later, a malignant lesion [22]. Recent evidence suggests that the dominant origin of HGS-EOC is the fimbrial epithelium, unlike low-grade serous, endometrioid, mucinous, or clear-cell ovarian cancer that arise from ovarian surface epithelium [20,21,22]. 

The stage and histological subtype of EOC are essential considerations to optimize the patients’ treatment because therapeutical success is poor when the cancer is detected in advanced stages. For instance, the 5-year relative survival rate in patients with invasive EOC is only 31%, while in localized EOC, it is 93% [23]. Moreover, studies have showed that survival rates of patients with EOC depend on the histological type. For example, patients with type II EOC had a significantly higher incidence of advanced disease (FIGO stages III/IV) than type I patients (79.8% vs. 38% respectively) being the overall survival and progression-free survival significantly higher in patients with type I tumours [24]. In addition, some studies have found a markedly higher mortality in patients with advanced mucinous and clear cell carcinoma, compared to higher survival rates of patients with HGS-EOC and endometrioid subtypes [25,26]. 

Current knowledge indicates that molecular characteristics of each patient with EOC should be considered to choose the best existing treatment, and personalized medicine should be considered in patients with EOC in the next future.

## 3. Molecular Characteristics of Epithelial Ovarian Cancer 

The ovary is a complex organ, which involves cyclic changes in endocrine, inflammatory, and nervous components. These characteristics influence not only the pathogenesis of EOC, but also its molecular heterogenicity and, potentially, the therapeutic response of these patients. 

### 3.1. SOMATIC and Germinal Mutations

Type I tumours are more genetically stable than type II tumours. Somatic mutations usually found in type I tumours include the genes of the RAF kinase family, Kirsten rat sarcoma virus protein (KRAS), beta-catenin, phosphatase and tensin homolog (PTEN), transforming growth factor-beta receptor II (TGF-β RII), phosphatidylinositol-4,5-bisphosphate 3-kinase subunit alpha (PIK3CA), and AT-rich interactive domain-containing protein 1A (ARID1A) [12,27]. 

In contrast, type II tumours rarely display the mutations found in type I tumours. They are chromosomally unstable and TP53 mutations are frequent (96% of HGS-EOC) [28]. Other recurrent mutations in type II tumours affect retinoblastoma protein (RB) and Notch signalling pathway [28]. Less frequent but not less important are genetic alterations in breast cancer type 1 susceptibility proteins (BRCA), because mutation or inactivation of BRCA genes and its downstream genes (via promoter methylation) occurs in up to 40–50% of HGS-EOC [19]. It is important to note that characteristic mutations of type I tumours could be found in type II tumours; however, these molecular changes are rarely important drivers in type II tumours [28] (Figure 1). 

Recent advances in EOC therapy have been modest, with few therapeutic options that significantly improve patients´ survival. A better knowledge of molecular characteristics of EOC requires further development of molecular-targeted therapies, which are just being explored in this neoplasm. 

### 3.2. Angiogenesis in Ovarian Cancer

Angiogenesis, or the generation of new blood vessels from other pre-existing, displays a considerable role in EOC [29,30]. In cancer cells, angiogenesis is enhanced to ensure the oxygen and nutrients supply, allowing tumoral growth and its dissemination. Many humoral factors are secreted by EOC cells to reach endothelial cells during tumour angiogenesis, promoting its proliferation, migration, and differentiation [30]. One of the most studied angiogenic factors is vascular endothelial growth factor (VEGF), which is largely produced and secreted by EOC cells [31,32,33]. This knowledge was used to develop the drug bevacizumab (Avastin), a monoclonal antibody against VEGF, which is currently used to treat EOC in advanced stages [3], a topic that will be discussed in detail later. 

Besides VEGF, other important pro-angiogenic molecules produced by EOC cells are neurotrophins. They display important functions in the nervous system and are involved in the correct ovarian performance [34,35]. Nerve growth factor (NGF) and brain-derived neurotrophic factor (BDNF) act as direct and indirect angiogenic factors, and their importance in EOC angiogenesis has been studied by several researchers [36,37,38]. Additionally, EOC produces a broad range of other angiogenic molecules, including placental growth factor (PlGF) [39,40], fibroblast growth factors (FGF) [41,42,43,44], platelet-derived growth factor (PDGF) [42,45] and angiopoietins [46]. 

Current studies have shown that the effects of angiogenic factors are not limited to endothelial cells, but they also produce autocrine stimulation in EOC cells [45,46,47]. The expression pattern of angiogenic factors could differ among histological types and significantly influences the progression-free survival and therapy response of patients with EOC [48], which should be considered for cancer treatment.

## 4. Current Therapies for Epithelial Ovarian Cancer 

### 4.1. First-Line Chemotherapy

The current treatment for EOC is debulking surgery. In advanced stages, the primary cytoreductive surgery followed by adjuvant chemotherapy remains the standard treatment for EOC [49,50]. Since the 80´s, the first-line chemotherapy has been based on platinum compounds, and during the 90’s taxanes were introduced, so depending on local guidelines, the standard chemotherapy is based only on platinum compounds or its combination especially in platinum-refractory or platinum-resistant patients [51,52]. In women with optimally debulked EOC (who did not receive the neoadjuvant treatment), adjuvant chemotherapy is not considered a reasonable option [51]. Intraperitoneal chemotherapy was introduced in the last decade, and some clinical trials have shown an advantage over intravenous administration, improving patients’ survival and tolerability [53,54,55].

Although complete remission is generally reached, most tumours will recur within two years, and rapid emergence of resistance to chemotherapy is observed [56]. The mechanisms responsible for chemoresistance to cisplatin and paclitaxel are diverse and include altered expression of membrane transporters, drug inactivation/detoxification, and resistance to cell death, among others [57,58]. However, beyond this acquired resistance, a plausible explanation for this phenomenon is the multiplication of a population of ovarian cancer tumour-initiating cells or stem-like cells, possibly originating from the ovary’s hilum region, that is proposed to initiate primary tumor growth, metastasis, and relapse of disease, but also for the development of chemoresistance [59,60]. 

To overcome platinum resistance, the United States food and drug administration (U.S. FDA) approved in 2006 the use of gemcitabine in combination with carboplatin to treat women with advanced ovarian cancer that relapsed at least six months after initial therapy [61,62]. Gemcitabine is a synthetic nucleoside inhibitor that increases the accumulation of cisplatin lesions producing cytotoxic synergy [63,64,65], enhancing the response to cisplatin treatment.

### 4.2. PARP Inhibitors 

The poly adenosine diphosphate-ribose polymerases (PARP) are essential enzymes involved in most cellular processes, including cell stress response, chromatin remodelling, DNA repair, and apoptosis [66,67,68]. They work by stabilizing PARP enzymes, inhibiting their activity, which prevents DNA repair and leads to cell death [68,69]. The four most used PARP inhibitors are olaparib (Lynparza), niraparib (Zejula), rucaparib (Rubraca) and talazoparib (Talzenna) [68]. These drugs are targeted agents for EOC with somatic or germinal mutations of BRCA1/2 genes or other genes that produce homologous recombination deficiency (HRD). HGS-EOC, the most common histological type of EOC, is characterized by frequent genetic and epigenetic alterations that produce HRD, most commonly BRCA1 and BRCA2 genes [28]. HRD is present in around a third of HGS-EOC, producing an aggressive phenotype [70,71,72].

The PARP inhibitor olaparib is the most studied in the context of cancer and was approved by the U.S. FDA as maintenance therapy for patients with EOC who have a partial or complete response to chemotherapy and have BRCA1/2 mutations [73]. Similarly, the U.S. FDA approved rucaparib as a single agent for treating relapsed ovarian cancer with mutations in BRCA genes in patients who had received two or more lines of chemotherapy [74]. Unlike the other PARP inhibitors, veliparib has not shown anti-proliferative activity, but radio and chemo-sensitizing effects were reported in cancer cells [75]. 

Regarding PARP inhibitors’ progress in EOC, Table 1 summarizes the most recent studies. It stands out an international clinical trial that evaluated carboplatin, paclitaxel, and veliparib induction therapy followed by veliparib maintenance therapy in patients with HGS-EOC, which showed a significantly longer progression-free survival than carboplatin plus paclitaxel induction therapy alone [76], suggesting that veliparib in association with chemotherapy or radiotherapy, could be used as chemosensitivity agent in HGS-EOC.

Additionally, PARP inhibitors have been studied as a single agent after chemotherapy or in combination with molecular-target agents. For instance, niraparib, which was approved in 2019 by U.S. FDA [77] has been tested in combination with pembrolizumab, an antibody against the programmed cell death receptor 1 (anti-PD1) in patients with platinum-resistant ovarian cancer. The use of niraparib plus the PD-1 inhibitor showed promising anti-tumoral activity in these patients, and importantly, responses of cancer patients without BRCA mutations (non-HRD) were higher than expected with either agent as monotherapy [78], which extend the groups of patients with the potential benefit of the use of PARP inhibitors. 

Other interesting associations studied include PARP inhibitors combined with anti-angiogenic therapy, such as cediranib (VEGF receptor inhibitor) or bevacizumab [79,80,81] as described in Table 1. 

**Table 1 biomedicines-10-00077-t001:** Summary of latest studies performed with PARP inhibitors in ovarian cancer patients.

Drugs	Study and Patients	Main Findings	Ref.
Chemotherapy in combination with veliparib (ABT-888) and as maintenance therapy	Phase III study. Advanced HGS-EOC ^1^	Veliparib increased progression-free survival compared to chemotherapy therapy alone in the HRD ^2^ cohort	[76]
Niraparib (Zejula) and pembrolizumab	Phase II study. Recurrent, platinum-resistant ovarian cancer	Responses of patients non-HRD were higher than expected either agent as monotherapy	[78]
Cediranib and olaparib (Lynparza)	Phase II/III study. Recurrent platinum-sensitive HGS-EOC	Drugs improved progression-free survival in patients with BRCA1/2 mutations	[79,80]
Chemotherapy with bevacizumab and olaparib (Lynparza) as maintenance therapy	Phase III study. Advanced HGS and endometroid EOC	The addition of olaparib increased progression-free survival in patients with HRD-positive tumours	[81]
Niraparib (Zejula) as maintenance therapy	Phase III study. Platinum-sensitive, recurrent ovarian cancer	Increase of progression-free survival in patients with or without BRCA mutations.	[82]
Olaparib (Lynparza) as maintenance treatment	platinum-sensitive relapsed ovarian cancer	Increased median overall survival of patients with BRCA mutations	[83]

^1^ HGS-EOC: high-grade serous epithelial ovarian cancer. ^2^ HRD: homologous recombination deficiency.

Even though BRCA mutations are the best predictors of the efficacy of PARP inhibitors, these drugs have shown positive effects in patients without BRCA mutations or non-HRD, which suggests that the use of PARP inhibitors may be extended to HGS-EOC, independently of the presence of HRD. 

### 4.3. Anti-Angiogenic Therapy (Bevacizumab)

Since exacerbated angiogenesis is a crucial characteristic of EOC cells and VEGF-A is the most expressed angiogenic factor in ovarian tumours, a therapy based on a human monoclonal antibody against VEGF-A seems promissory for EOC treatment. In this context, the U.S. FDA approved in 2018 the use of bevacizumab (Avastin), a monoclonal antibody against VEGF-A as first-line treatment for epithelial ovarian, fallopian tube, or primary peritoneal cancer stage III or IV in combination with carboplatin and paclitaxel [3]. Similarly, the European Commission approves the use of bevacizumab in combination with standard chemotherapy as a treatment for women with the first recurrence of platinum-sensitive ovarian cancer and first-line chemotherapy following surgery in women with advanced ovarian cancer [84].

Bevacizumab has been studied in clinical trials that include patients with recurrent platinum-sensitive and platinum-resistant ovarian cancer. In the first case, OCEANS trial showed that the addition of bevacizumab to gemcitabine and carboplatin therapy improved the progression-free survival of patients [85]. Alike, AURELIA trial, that studied the combination of bevacizumab to chemotherapy in patients with platinum-resistant EOC, shows that the benefit of bevacizumab therapy in advanced EOC was modest, increasing in a few months the progression-free survival of patients and without significant changes in the overall survival of intervened patients [86]. Similar results were obtained in the trial GOG-0218, a phase III randomized trial of bevacizumab in women with newly diagnosed ovarian cancer, that showed no survival differences for patients who received bevacizumab compared with chemotherapy alone [87]. 

To a better understanding of the real benefit of using bevacizumab in ovarian cancer patients, a meta-analysis that included 7 studies with patients with advanced ovarian cancer was performed [88]. This study concluded that bevacizumab treatment increased progression-free survival in patients with both advanced and recurrent disease, but its use was associated with an increase of overall survival only in patients with recurrent disease [88]. Although bevacizumab therapy is extended to many countries and is considered one of the greatest advances in the treatment of ovarian cancer, some researchers consider that this therapy could be not cost-effective [89,90].

The inhibition of VEGF-mediated signalling leads to tumour vasculature normalization, improving chemotherapy delivery which results in increased tumour toxicity and a decreased formation of ascites fluid [91,92]. This knowledge suggests that selecting appropriate patients for bevacizumab treatment could contribute to improve therapeutic efficacy. Results from the trial GOG 0218 show that patients with ascites treated with bevacizumab had a significant improvement in progression-free and overall survival, which was not observed in patients without ascites, suggesting that ascites predicts treatment benefit of bevacizumab in patients with advanced EOC [93]. Similarly, a phase I study in patients with diverse cancers tested intraperitoneal bevacizumab for treating refractory malignant ascites. Preliminary results of this study showed that bevacizumab exhibited short-term anti-tumoral efficacy and palliated symptoms [94]. Given that over one-third of women with ovarian cancer will develop ascites [95], bevacizumab therapy should be considered in this subgroup of patients.

Further, it is believed that polymorphisms of several genes involved in angiogenic pathways could be associated with the efficacy of bevacizumab in cancer treatment [96], including genetic variants in the renin-angiotensin system [97]. In ovarian cancer patients, it was described that a specific polymorphism of interleukin 8 (IL-8) may predict the response to bevacizumab-based chemotherapy [98]. Another recent study suggests that patients with ovarian cancer that express low levels of the tumour-suppressor micro-RNA-25 (miR-25) will have significant benefit from bevacizumab treatment in terms of progression-free survival and overall survival [99]. 

Currently, other anti-angiogenic compounds are being tested to improve the response of EOC treatment (Figure 2). For instance, aflibercept is an antiangiogenic soluble fusion protein that acts as a “VEGF trap” and inhibits VEGF-A and VEGF-B, as well as PlGF signaling [100,101]. Two clinical trials show that aflibercept was effective in controlling malignant ascites with a safety profile [102,103], even though the drug shows a significant risk of fatal bowel perforation of patients with very advanced cancer, which suggests that the benefit-risk balance should be discussed with each patient.

Even though VEGF is the most studied angiogenic factor in ovarian cancer, other important pro-angiogenic molecules could be upregulated in response to anti-VEGF therapy. This theory could explain the failure or resistance to anti-angiogenic therapy in some patients.

## 5. Inhibitors of Other Angiogenic Factors That Are Being Tested in EOC

Apart from VEGF, EOC cells could yield and release several other angiogenic factors such as angiopoietins, neurotrophins, and PDGF, among others [30,38,39,40,46,104]. Based on this knowledge, specific inhibitors of these molecules (summarized in Table 2) have been developed. An example is trebananib (AMG-386), an angiopoietin 1 and 2 neutralizing chimeric protein (peptibody). It binds angiopoietins thereby preventing the interaction with their cell surface receptors, inhibiting angiogenesis and tumoral growth [105]. Trebananib has been tested in combination with carboplatin and paclitaxel as first-line treatment for advanced ovarian cancer in the clinical trial TRINOVA-1/3. Unfortunately, the trial shows that the addition of trebananib to standard therapy was minimally effective and did not improve the progression-free survival of patients [106,107]. 

Because anti-angiogenic therapy is a key point in EOC and most of the angiogenic and growth factors have in common the activation of tyrosine kinase (RK)-mediated signalling, selective or non-selective TK inhibitors have emerged as alternative drugs in the context of EOC therapy. For instance, sorafenib (Nexavar) is a multiple protein kinase inhibitor that decreases the signalling of VEGF and PDGF receptors [111,112] and is approved by the U.S. FDA for the treatment of patients with advanced renal cell carcinoma and unresectable hepatocellular carcinoma [113]. Unfortunately, a clinical trial shows that sorafenib had only modest anti-tumoral activity and substantial toxicity in patients with recurrent ovarian cancer [114], but this combination with other drugs shows better results. For instance, using sorafenib in combination with topotecan (topoisomerase inhibitor) as maintenance therapy improves the progression-free survival of patients with platinum-resistant ovarian cancer [108]. Similarly, a phase II study showed a clinical benefit of the combination of sorafenib and bevacizumab in bevacizumab-naïve EOC patients who were heavily pre-treated with platinum. However, the study highlights the importance of close monitoring and dose modifications in these patients due to toxicity of the drug combination [109], which puts into question the effectiveness and security of the use of tyrosine kinase inhibitors in EOC.

## 6. Anti-Neurotrophins Therapies as Possible New Approaches in EOC

Neurotrophins and their receptors are a group of molecules whose importance in the nervous system is widely known. In the last decades, their contribution to the homeostasis of other non-neuronal tissues has been described. For instance, neurotrophins dysregulation has been reported in some ovarian pathologies such as polycystic ovarian syndrome [115,116] and EOC [30,38]. 

Neurotrophins and their receptors display a crucial role in the progression of EOC, acting as autocrine growth factors and angiogenic factors [36,37,38,117]. The most studied neurotrophins, NGF and BDNF, bind their high-affinity tropomyosin receptor kinases (TRK) A and B, respectively [118]. In EOC, their upregulation has been associated with poor survival rates [37,119]. Therefore, neurotrophins and their receptors have been proposed as potential therapeutic targets in EOC.

TRK fusion is a phenomenon present in diverse kinds of cancer [120]. In EOC, the presence of TRK fusions in patients’ biopsies is not documented, but it is assumed, because of the critical contribution of neurotrophins and their receptors to EOC progression. Based on this, two clinical trials are testing pan-TRK inhibitors in patients with neoplasms, including EOC (NCT02568267 and NCT03215511). These inhibitors are small molecules that bind to TRK receptors, prevent neurotrophins-TRK interaction and, therefore, TRK activation [121]. In 2018 and 2019, the U.S. FDA approved larotrectinib (Vitrakvi) and entrectinib (Rozlytrek) respectively for the treatment of adult and paediatric patients with solid tumours that have TRK gene fusions [122,123]. Another recently developed TRK inhibitor is loxo-195 (Selitrectinib), a second-generation drug that overcomes the acquired resistance to first-generation TRK-inhibitors [124]. 

A recent report on the use of TRK-inhibitors in EOC shows that entrectinib induced durable and clinically meaningful responses in patients with TRK fusion-positive solid tumours, being well tolerated with a manageable safety profile. However, only 2% of these participants had ovarian cancer [110], so the usefulness of TRK inhibitors as a complementary therapy in EOC needs further study. 

## 7. Immune Checkpoint Inhibitors as an Alternative for Ovarian Cancer Treatment

Cells with tumour potential are constantly produced in the human body but the immune system oversees their elimination. Immune checkpoints are modulators of immune response which is crucial for self-tolerance, preventing autoimmunity and the shutdown of exacerbated responses. In this context, regulatory T cells (Treg) and inhibitory surface molecules, including cytotoxic T lymphocyte-associated protein 4 (CTLA4), programmed cell death receptor 1 (PD1) and its ligand (PD-L1), are induced during immune responses and represent immune checkpoints [125,126]. Cancer cells manipulate these mechanisms to avoid the immune response, preventing their elimination [127]. 

Different inhibitors of immune checkpoints have been developed to treat solid cancers and some of them have been tested in patients with EOC. In November 2021, there were more than 120 clinical trials using checkpoint inhibitors in ovarian cancer patients inscribed in clinicaltrials.gov (accessed on 7 December 2021), whose results are summarized in Table 3. These studies concluded that the expression of PD-L1 in EOC cells, the histotype, and previous treatment are associated with the success of immune therapies.

The phase Ib KEYNOTE-028 study tested pembrolizumab in 26 patients with PDL1-positive advanced ovarian cancer, showing modest but durable anti-tumor activity with an overall response rate (ORR) of 11.5% [130]. Phase II of this study (KEYNOTE-100 cohort) tested the same drug in 376 patients with recurrent ovarian cancer [131]. The study showed a higher ORR in HGS-EOC and clear cell ovarian cancer subgroups, a better response in patients with five or more lines of previous treatment, and an increased ORR in patients with a high presence of PDL1 (combined positive score > 10). In addition, the study showed that 8% of patients had a complete or partial response to pembrolizumab monotherapy, while progressive disease was reported in 57.2% of patients [131].

Among other current immunologic therapies, a clinical trial intends to test the drug ipilimumab (anti- CTLA-4 antibody) in ovarian cancer patients (NCT00060372). Although the preliminary reports are available only for patients with hematopoietic malignancies, they showed encouraging results [135]. On the other hand, given that ovarian cancers express high levels of mesothelin [136], a clinical trial is testing a monoclonal antibody anti-mesothelin (ABBV-428) as monotherapy in patients with several cancers, including ovarian cancer (NCT02955251). ABBV-428 targets mesothelin via a C-terminal single-chain variable fragment flanking Fc-modified human IgG1 and CD40 via an N-terminal single-chain variable fragment, which produces the activation of CD40 [137]. CD40 acts via ligation on antigen-presenting cells, stimulating T-cell activation and proliferation [138]. Unfortunately, the first results of this trial showed minimal clinical activity in a small cohort of patients with advanced ovarian cancer [137].

Another interesting approach regarding immunotherapies is the development of cell-specific vaccines. For instance, DPX-Survivac (DepoVax) is a vaccine that generates a tumour-specific immune response, particularly by cells that express the protein survivin, using survivin HLA class I peptides [139]. Because ovarian cancer is one of the neoplasms that express higher amounts of this protein, a clinical trial tested the combination of DPX-Survivac, a low dose of cyclophosphamide and epacadostat, an inhibitor of indoleamine 2,3-dioxygenase-1 (IDO1) which may reverse tumour-associated immune suppression (NCT02785250). Preliminary results of this study showed encouraging results in 3 of 10 patients, 2 of them with a disease control for more than 12 months [140].

A current perspective using immune checkpoint inhibitors is the combination of these drugs with anti-angiogenic therapies [141] and with PARP inhibitors [142] which could decrease primary resistance, improving therapy results. This is because PARP inhibitors activate the generation of type I interferon response, which upregulates chemokines that leads to T-cell recruitment and to PD-L1 upregulation on cancer cells [143,144]. In addition, BRCA dysfunction increases T-cell recruitment to the tumour site and increases the expression of immune response genes as PD1 and PD-L1 [145,146]. The combined use of immune checkpoint inhibitors with PARP inhibitors promotes the sensibilization to the second ones in breast cancer cell lines [147] and greater anti-tumoral activity than either drug alone, suggesting that this combination could be a rational strategy. In ovarian cancer patients, the combination of olaparib and durvalumab was tested in heavily pretreated patients, showing clinical activity in patients without BRCA mutations [148]. Another phase I study tested olaparib and tremelimumab in women with heavily pre-treated and recurrent BRCA-associated ovarian cancer. Preliminary results of this study showed acceptable tolerability and therapeutic effect [149].

## 8. Drug Repurposing for Complementary Treatment for Ovarian Cancer

Drug repurposing is the process of identifying new therapeutic uses for existing or available drugs. It is an effective strategy in discovering molecules with new therapeutic implications [150,151]. Because existing drugs have studies of pharmacokinetics and safety in humans, the approval for further therapeutical use is shorter than the conventional development of a new drug, which could be especially beneficial in lower-income countries. The increased knowledge about the mechanisms involved in the progression of EOC has promoted several studies of repurposed drugs as possible complementary therapies (summarized in Table 4 and Figure 3), and most of them are addressed in the following sections.

### 8.1. Autophagy Inhibitors (Antiparasitic Drugs)

Autophagy is an evolutionary form of self-digestion whose pathways are involved in protein and organelle degradation, and its imbalance is observed in several human diseases, such as EOC [169,170,171]. Autophagy increases during nutrient and growth factor deprivation, endoplasmic reticulum stress, development, or accumulation of protein aggregates [169]. 

The tumour microenvironment has exceptionally stressful conditions, including hypoxia and nutrient deprivation, and autophagy allows cancer cells to survive under these metabolic stress conditions [172]. In cancer cells, it is described that autophagy plays a dual role. On the one hand, it promotes cell death and cell cycle arrest, which usually prevents tumour development [173,174,175]. On the other hand, autophagy operates as a mechanism for tumour adaptation, reducing damaged cellular parts, recycling intracellular components to supply metabolic substrates, and maintaining cellular homeostasis, promoting tumour survival and growth in advanced cancers [176,177]. In addition, it is involved in the increase of resistance of anticancer drugs and other essential processes, including oxidative stress, inflammation and modulates tumour immunology [178,179,180]. In EOC cells, the increase of autophagy leads to cisplatin resistance, and their inhibition mediates cisplatin sensitivity [181,182]. Therefore, it is believed that autophagy inhibition can re-sensitize resistant cancer cells to chemotherapy and increase its cytotoxicity.

Chloroquine and its derivatives are common autophagy inhibitors. These compounds could be accumulated in intercellular acid vesicles as lysosomes, thereby inhibiting lysosome–autophagosome fusion [183]. In 2018, a case report showed that a 60-year-old woman with advanced and not resected intra-abdominal EOC achieved a complete response to chemotherapy after receiving hydroxychloroquine and quinacrine [184]. This finding fostered the interest of several researchers in the anti-tumoral effects of antimalarials in ovarian cancer. 

In vitro experiments have shown that chloroquine reverses cisplatin resistance in EOC cells, producing autophagy inhibition and lethal DNA damage by inducing p21WAF1/CIP1 expression [152]. Besides, autophagy inhibitors have been tested in combination with other anti-tumoral compounds. In glioblastoma cells, hydroxychloroquine potentiates the anti-cancer effect of bevacizumab [185]. These findings encouraged a phase I/II trial that assesses hydroxychloroquine and itraconazole in women with advanced platinum-resistant EOC (HYDRA-01). The study shows that even if a high presence of autophagy markers was detected in 30% of patients, the drug combination did not show patient benefits [153]. It is important to highlight that this study included a heavily pre-treated platinum-resistant EOC population; therefore, the effect of autophagy inhibitors in EOC patients without previous treatment or in less advanced stages of the disease is still unknown. 

Another broad-spectrum antiparasitic drug whose anti-tumoral effects are mediated by the increases of autophagy is ivermectin [186]. In ovarian cancer cells, ivermectin synergistically suppresses tumour growth in combination with cisplatin [155] or paclitaxel [154], and its anticancer mechanism involves the modulation of long non-coding RNAs with multiple targets [187]. However, these effects have not been tested in humans yet.

### 8.2. Lipid-Lowering Medications

Statins are a group of drugs widely used to reduce cholesterol biosynthesis inhibiting the enzyme HMG-CoA reductase [188]. A Danish study performed in 2012 showed that statin use in patients with cancer was associated with reducing cancer-related mortality [189]. In agreement with this, a meta-analysis conducted with different studies of statins and gynaecological cancers showed that the use of these drugs was inversely associated with ovarian cancer risk [190]. 

However, some studies showed no association between the use of statins and reduced risk of ovarian cancer [191,192]. Therefore, the evidence suggests a protective role of statins in ovarian cancer, but this is not conclusive. To elucidate this controversy, researchers have performed studies using EOC cells as well as more detailed analyses of published studies. In vitro experiments have shown that statins (lovastatin and atorvastatin) inhibit cell proliferation, suppress anchorage-independent growth, induce apoptosis, autophagy, cellular stress, and cell cycle arrest [193,194,195] in different EOC cell lines. Currently, there is a clinical trial recruiting patients with platinum-sensitive ovarian cancer to test the effect of simvastatin in the progression of the disease (NCT04457089).

The conflicting evidence about the positive effects of statin in ovarian cancer patients could be explained by a differential effect according to the histotype of ovarian carcinoma. A recent systematic review [156] that included 9 studies of statins in ovarian cancer showed a new perspective. The study concluded that the use of hydrophilic statins was associated with a decrease of risk of ovarian cancer (unlike hydrophobic, whose use increased the risk), particularly in mucinous and endometroid subtypes, a higher protective effect with long term use of statins (>5 years) and a most significant benefit with the combination of statins with other drugs, as salicylic acid.

### 8.3. Bisphosphonates 

Bisphosphonates are pharmacological agents used against osteoclast-mediated bone loss [196]. Nitrogen-containing bisphosphonates (second generation of these drugs) inhibit the activity of farnesyl pyrophosphate synthase, a key regulatory enzyme in the mevalonic acid pathway. This substrate is critical to producing sterols and isoprenoid lipids whose deficiency produces osteoclast apoptosis [197,198]. 

In vitro studies have shown that pamidronate, incadronate, alendronate, risedronate and zoledronate had direct inhibitory effects on cell proliferation of several ovarian cancer cell lines [157,199]. However, these results are not consistent with a few studies performed with ovarian cancer patients. A systematic revision [157] studied the relationship between the use of bisphosphonate and the risk of endometrial and ovarian cancer. The study evidenced that the use of bisphosphonates for more than one year was associated with a reduced risk of endometrial cancer, but ovarian cancer risk remains unchanged. However, this meta-analysis included only four studies with ovarian cancer patients. 

On the other hand, another study evaluated the effect of bisphosphonates and lipid-lowering medications in EOC cells. Results showed that the treatment with pitavastatin and zoledronate displayed additive and synergistic anti-proliferative effects on most ovarian cancer cell lines [157]. Another study in EOC cell lines showed that tumour-promoting cytokines and mediators, such as transforming growth factor (TGF)-β1, VEGF, interleukin (IL)-8, and IL-6, were suppressed up to 90% after the treatment with statins and zoledronate [158]. 

Because ovarian cancer affects women in the age of 65 years and older more frequently than younger [5,200] and osteoporosis is the most prevalent disease in menopausal women [201,202], the use of bisphosphonates in association with other known medications as statins could be an interesting field that should continue to be investigated. 

### 8.4. Pro-Oxidative Drugs

In the context of repurposing drugs, some pro-oxidative agents have shown anti-tumoral activity in EOC cells. An evident example is disulfiram, one of three drugs approved by the U.S. FDA to treat alcohol dependence, whose mechanism involves the irreversible inhibition of aldehyde dehydrogenase (ALDH1A1) [203]. ALDH1A1 is not only a hepatic enzyme implicated in the major oxidative pathway of alcohol metabolism; it is also considered a stem cell marker that promotes epithelial-mesenchymal transition (EMT) progress in EOC cells [204,205]. The activity of ALDH1A1 is significantly higher in taxane- and platinum-resistant cell lines, and notably, the presence of ALDH1A1-positive cells is negatively correlated with progression-free survival in HGS-EOC patients [159]. Therefore, inhibitors of this enzyme could be helpful in the context of EOC. 

In ovarian cancer cells, the treatment with disulfiram produces dose and time-dependent cytotoxic effects, enhancing cisplatin-induced apoptosis, and consequentially their association with the cofactor copper increases intracellular ROS levels, triggering apoptosis of ovarian cancer with stem cell phenotype [160]. In the same way, it was reported that disulfiram caused irreversible cell damage in EOC cells by redox-related proteotoxicity associated with induction of heat shock proteins HSP70, HSP40, and HSP32 [206].

Another pro-oxidative compound with anti-tumoral effects in EOC cells is arsenic trioxide (As_2_O_3_). The treatment with this compound is used as first-line and consolidation/maintenance treatments in haematological pathologies such as promyelocytic myeloid leukaemia [207]. Arsenic trioxide has been tested on ovarian cancer cell lines in both in vitro and animal models [208]. It can sensitize ovarian cancer cells to PARP inhibitors and cisplatin resistance [161], it has anti-angiogenic and antiproliferative activities by decreasing the expression of VEGFA and topoisomerase II, respectively [209,210], and produces cell growth inhibition and increase of apoptosis in adherent and suspension ovarian cancer cells, along with a synergism with cisplatin and paclitaxel treatment [162,211,212]. These findings encouraged two clinical trials registered in ClinicalTrials.gov (accessed on 7 December 2021) database testing the effect of arsenic trioxide in platinum resistance relapsed ovarian cancer (NCT04518501) and recurrent and metastatic ovarian cancer with P53 mutation (NCT04489706). 

### 8.5. mTOR Inhibitors

One of the most studied signalling pathways in cancer is the mechanistic target of rapamycin (mTOR), two protein complexes that regulate cell growth, survival, metabolism, nutrient input, drug resistance, and immunity [213,214]. One example of mTOR inhibitor is itraconazole, a broad-spectrum antifungal agent, that had anti-proliferative effects in EOC and endothelial cells [215]. Currently, there are two clinical trials studying itraconazole in the context of EOC. The first aims to evaluate the effects of itraconazole and tamoxifen in platinum-refractory/resistant or recurrent ovarian cancer (NCT03458221). The second is the previously mentioned study HYDRA-01 that tested the combination of hydroxychloroquine and itraconazole in women with heavily pre-treated platinum-resistant EOC. This study is the only one with results, which unfortunately did not show a patient benefit [153].

#### Biguanides 

Other widely studied mTOR inhibitors are biguanides, of which metformin, phenformin, and buformin are the main representants. Due to reports of lactic acidosis (the more serious adverse effect), phenformin and buformin were withdrawn from clinical use in most countries during the 1970s [216]. Metformin, which has a much lower risk of lactic acidosis, is the most used biguanide for treating type 2 diabetes and metabolic disorders [216]. In the context of EOC, both metformin and phenformin have shown anti-proliferative effects in-vitro and in-vivo [217,218], but most studies have been conducted using metformin.

In humans, metformin intake has been associated with decreased cancer incidence and mortality in type 2 diabetic patients [219], including women with EOC [163,164]. Besides, in vitro studies have shown that metformin exerts multiple and pleiotropic anti-tumoral effects in EOC cells [218,220]. One of the most studied molecular targets of metformin is the adenosine monophosphate-activated protein kinase (AMPK), a key sensor of the energetic status of the cell [221]. AMPK activation inhibits mTOR complex 1 (mTORC1), impairing cancer cell survival, protein synthesis (which regulates cell proliferation and immune cell differentiation), and tumoral metabolism [222]. Although metformin is a known activator of AMPK, studies have reported that their anti-tumoral effects are dependent and independent of AMPK inhibition [223,224].

The treatment of HGS-EOC cells with metformin or phenformin decreases cell proliferation and produces changes in their cell metabolism, increasing glycolysis and inhibiting oxidative phosphorylation by alteration of mitochondrial shuttle metabolites [225,226]. In addition, studies have shown that metformin enhances cisplatin cytotoxicity in EOC cells [128] and produces chemo-sensitizing effects in cisplatin- and paclitaxel-resistant EOC cells [227]. 

Some targets of metformin in EOC cells include angiogenic factors that are overexpressed by tumoral cells such as VEGF or NGF; the proteins sterol regulatory element-binding protein 1 (SREBP) and acetyl-CoA carboxylase (ACC) (critical proteins involved in fatty acid synthesis), c-MYC transcription factor, cyclins, cell cycle regulators and EMT proteins [218,228]. In addition, metformin elicits anticancer effects through the sequential modulation of the endoribonuclease Dicer [229], an important component of microRNAs biogenesis, which is dysregulated in EOC, which is further discussed later.

Tumour cells grow in environments with scarce nutrients and oxygen, known as the tumour microenvironment [230]. The resulting microenvironments contribute to the development of cellular subpopulations with different metabolic characteristics. These populations include stem-like cells, which adapt to reduced oxygen availability, switching between glycolysis and oxidative phosphorylation as energy sources and metabolites [230]. Epidemiologic and preclinical studies suggest a selective anti-tumoral effect of metformin on stem-like EOC cells [231,232,233,234], which plays an essential role in chemoresistance and ovarian cancer recurrence [60,235]. In this context, a phase II trial evaluated the impact of metformin on stem-like EOC cells and carcinoma-associated mesenchymal stem cells in nondiabetic patients with advanced EOC [165]. Results showed that tumours from metformin-treated patients decrease cancer stem cells markers and increase sensitivity to cisplatin ex vivo. Additionally, metformin altered the methylation signature in carcinoma-associated mesenchymal stem cells, which prevented the chemoresistance mediated by these cells in vitro [236]. 

Biguanides have been tested with other antitumoral agents, such as PARP inhibitors. Phenformin and metformin have shown a synergistic effect with olaparib, reducing cell survival, tumorigenesis, and decreasing mesenchymal markers of drug-resistant ovarian cancer cells [237], so the association of metformin with other new anti-tumoral drugs could be an important research matter.

Because metformin has additional beneficial effects against cancer, including anti-inflammatory, anti-aging, and antithrombotic properties [238], it was proposed as a plausible complementary therapy to first-line treatments for EOC patients. Most in vitro studies have corroborated the anti-tumoral potential of this drug; however, few clinical trials have tested metformin in non-diabetic patients with EOC. Although there is much in-vitro evidence indicating that metformin could be helpful as an anti-tumoral drug, many questions remain to be answered, such as its anti-tumoral mechanism, dose, timing, and the optimal therapeutic window (before cytoreduction? during chemotherapy cycles? both?). It is relevant to continue studying the anti-tumoral effects of metformin in patients with earlier stages and without previous treatment, along with establishing the best dose and optimal therapeutic window that could benefit to EOC patients, as well as the combination with other possible drugs. 

### 8.6. Non-Steroidal Anti-Inflammatory Drugs (NSAIDs)

NSAIDs are widely used to relieve pain, reduce inflammation, and bring down a high temperature [239]. The primary mechanism of action of NSAIDs is the inhibition of the enzymes cyclooxygenases (COX) 1 and 2 and, therefore, prostaglandin (PG) synthesis [239]. Long-term use of NSAIDs has been associated with reduced incidence of several epithelial cancers [240,241,242]. A molecular explanation is that chronic inflammation promotes carcinogenesis by inducing proliferation, angiogenesis, metastasis, and chemotherapy resistance [243], so the use of NSAIDs could be an adequate possibility to improve cancer treatments targeting the inflammation.

In addition to the direct anti-tumoral effects of NSAIDs in cancer cells, their use could be beneficial as complementary use to chemotherapy. It is thought that the combination of cell death and PGE2 release due to cisplatin treatment results in a wound-like response and initiates a stem-like program, which favours cancer progression. In bladder cancer cells, the use of combined cisplatin and COX-2 inhibitor celecoxib prevents cisplatin resistance and restores cisplatin sensitivity in vivo and in vitro [166,167]. In neuroblastoma, diclofenac (a known NSAIDs drug) enhanced chemotherapy-induced apoptosis via upregulation of p53 [244]. Particularly in EOC, it is described that COX inhibitors increase paclitaxel sensitivity in taxane-resistant EOC cells [245]. 

Among NSAIDs, diclofenac is a relevant drug in cancer therapy. One significant footprint of human cancers is the metabolic switch which favours glycolytic pathways to obtain energy in the form of ATP (Warburg effect) [246]. Cancer cells increase glucose uptake, and as a result of an increased glycolysis rate, high concentrations of lactate are produced [247]. Lactate is considered an immunosuppressive metabolite [248], and under low glucose, tumour cells uptake and oxidize lactate, which means that lactate is used as an energetic source by cancer cells [247,249]. Research performed in a murine glioma model demonstrated that diclofenac (but not ibuprofen, another known NSAID), decreased lactate dehydrogenase A and lactate secretion. Additionally, it was described that diclofenac inhibited the uptake of lactate in colorectal cancer cells, being the most potent inhibitor among other NSAIDs [250]. 

This in vitro evidence suggests that NSAIDs could be helpful not only as anti-inflammatory agents, but also as inhibitors of cell metabolism, as the drug metformin.

## 9. Non-Coding RNA-Based Therapeutics for Ovarian Cancer

Non-coding RNAs are one of the latest emergent tools for targeting-cell therapy. Around 98% of the human genome corresponds to non-protein-coding sequences, and a substantial part of them are non-protein-coding RNA transcripts (ncRNAs) [251]. ncRNAs are RNA molecules that are not translated into proteins but have crucial cellular functions so that their dysregulation can lead to the development of different pathologies such as cancer [252]. The ncRNAs are classified according to their size into small non-coding RNAs and long non-coding RNAs (lncRNA), greater than 200 nucleotides [253].

Among the small ncRNAs, microRNAs (miRs) play a key role in the initiation and progression of ovarian cancer [254]. miRs are short sequences of nucleotides that bind messenger RNAs and interfere in protein translation [255], so they are post-transcriptional regulators. In cancer, there is a miR disbalance, with a predominance of oncomiRs (that inhibits the transcription of oncosupressor proteins) and a decrease of tumour-suppressor miRs, resulting in an increase of oncoproteins [255]. Because one miR regulates several messenger RNAs (and several different proteins) involved in critical tumoral processes such as proliferation, migration, invasion, and angiogenesis [254], the use of miR-based therapy has been attractive for most researchers. Among different miRs, dysregulation of Let-7, miR-200 family, miR-17-92, miR-21, miR-145, and miR-23b have been reported in ovarian cancer, being the first one of the most studied miR in the context of EOC [256]. miR-145 is an onco-suppressor miR that is downregulated in EOC biopsies and EOC cells. Studies have shown that overexpression of miR-145 decreases the cell proliferation, migration, invasion, and tumoral formation of EOC cells [257]. In addition, the role of this miR in tumoral metabolism has been documented. miR-145 negatively regulates the Warburg effect in bladder cancer cells [258] and inhibits the mitochondrial function of EOC cells [259], suggesting that its reinstatement could contribute to the energetic depletion of tumoral cells. Significantly, cisplatin mediates downregulation of miR-145 in cisplatin-resistant EOC cells [260] and upregulation of miR-145 sensitizes EOC cells to paclitaxel, suggesting a crucial role of miR-145 in chemotherapy resistance. Because miRs are “natural” molecules present in all bodies, with likely no adverse effects in other tissues [253], the re-establishment of levels of onco-suppressor miRs as miR-145 could be an interesting proposition as a future complementary treatment in EOC.

On the other hand, lncRNAs are also involved in miRs disbalance. They play an important role in gene expression and may even function as miRs sponges, which may cause a decrease in the effect of some miRs [261,262].

Metastasis-associated lung adenocarcinoma transcript 1 (MALAT1) is one of the best-described lncRNAs and its up-regulation has been associated with progression and chemoresistance in different types of cancers [263,264,265]. In ovarian cancer, MALAT is overexpressed in epithelial ovarian cancer tissues and cell lines, promoting proliferation and metastasis via the PI3K-AKT pathway [266,267]. Recent studies in prostatic cancer cells determined that MALAT1 promotes the proliferation, migration, and invasion of tumour cells by acting as a miR-145 sponge, suppressing its anti-tumoral activity, and the downregulation of MALAT1 increases miR-145 and decreases cell migration and invasion [268]. These observations led to the proposition of oncogenic sponge lncRNAs as possible therapeutic targets for cancer treatment, and their specific modulation could be an interesting approach in EOC.

## 10. New Methods of Drug Delivery (Nanomedicine)

For decades, the anti-tumoral drugs were chemically modified to increase their half-life and biodistribution. However, the tissue-specific drug delivery using nanotechnology (nanocarriers) is set to spread rapidly and gained much interest as a promising diagnostic and therapeutic strategy (Figure 4). 

Research groups have explored new methods of drug delivery, such as nanoparticles, in different ovarian cancer models, with the purpose to improve the effectiveness of chemotherapy [269]. In this context, nanocarriers that address chemotherapeutic drugs or new anti-tumoral biological molecules, such as miRs or small interference RNAs (siRNAs) have been tested in vivo. They could be directly delivered into target cancer cells, resulting in enhanced therapeutic impact with less toxicity, biocompatibility, good biodegradability and increased therapeutic impact than free drugs, with fewer side-effects [269,270,271].

Targeted drug delivery has become a new paradigm in cancer therapy [271] using a range of nanomaterials based on organic, inorganic, lipid, or glycan compounds and synthetic polymers [272]. For instance, nanoparticles in base to hyaluronic acid encapsulating both paclitaxel and focal adhesion kinase (FAK) siRNA were developed as a selective delivery system against chemoresistant EOC cells. This nanoformulation strongly decreased the tumoral growth of EOC xenografts and patient-derivate xenografts [273]. Similarly, iron-oxide nanoparticles (Fe_2_O_3_) have emerged as one of the extensively utilized nanostructures in different models of EOC, because of their superparamagnetic features, which enables their accumulation in animal tumours, and appear as a candidate for properties such as antioxidant, antibiofilm, antimicrobial, and antitumoral activities [274,275,276]. A recent work showed that Fe_2_O_3_ nanoparticles display cytotoxic activity in metastatic ovarian teratocarcinoma cells by augmenting the level of reactive oxygen species (ROS), destabilizing mitochondrial membrane, and enhancing of programmed cell death [277]. Other nanocarrier examples include cationic liposomes with polyethylene glycol and glycolic acid-based nanoparticles, which may incorporate chemotherapeutics such as paclitaxel or doxorubicin and small interference RNAs (siRNA) producing synergistic antitumoral effects in in vitro and in vivo models of EOC cells [278,279,280].

Another recent approach of molecular agents that are tested in several ovarian cancer models is the use of extracellular vesicles (EVs). They are continuously produced and released by cells and contain proteins, messenger RNA, and miRs that can be transferred to another cell and become functional in the new location [281]. EVs play an important role in cancer diagnosis [282] or prognosis [283] and drug resistance [284] and it is proposed that they could be used as a delivery vehicle of chemotherapeutics [285]. Two main kinds of EVs are produced to carrier and deliver anti-tumoral therapies: EVs-based semisynthetic nanoparticles and EVs mimetic nanoparticles. The first is naturally isolated EVs with surface and membrane modifications; while EVs mimetic nanoparticles are artificial structures [286] that could efficiently encapsulate a different kind of anti-tumoral molecules. Among EVs, exosomes, a subtype of EVs with a diameter less than 150 nm, are fully considered as possible nanocarriers due to their size and the capacity to carry a wide variety of molecular cargos [281].

A recent work shows that stem cell-derived EVs can release the encapsulated miR-424, suppressing tumorigenesis and angiogenesis of ovarian tumours in vivo [287]. Not only biological compound could be delivered by EVs, but also hydrophobic or toxic chemotherapeutics. For instance, exosomes from tumour cells (modified with biomimetic porous silicon nanoparticles) were used as drug nanocarriers for targeting doxorubicin [288]. This nanoformulation enhanced tumour accumulation, extravasation from blood vessels, and penetration into deep tumour parenchyma of doxorubicin in a model of mouse hepatocellular carcinoma [288]. In ovarian cancer cells, researchers have studied the encapsulation of triptolide (a natural product isolated from the Chinese herbal) in exosomes from EOC cells, obtaining a high drug encapsulation efficiency and uptake by SKOV3 cells [289]. In vivo, this exosome formulation showed more potent inhibition of tumour growth and less toxic effects on the liver and spleen than free triptolide extract, concluding that exosome delivery could be helpful to address and decrease the side effects of antitumoral agents [289]. 

Inspired by the self-assembly that occurs in cells and exosomes, researchers designed a biomimetic lipid/dextran hybrid nanocarrier loaded with a siRNA for multidrug resistance protein 1 (MDR1) and the drug paclitaxel, which reversed the chemoresistance in both in vitro and in vivo models of EOC cells [168]. These synthetic structures based on exosomes could be a promising strategy for EOC treatment because they can overcome some difficulties inherent to the natural purification of exosomes, such as the problematic obtention and variable retrieval of extracts, which should be studied for the following years.

## 11. Main Conclusions

EOC is a neoplasm with high mortality and late diagnosis. Because of its molecular heterogenicity and different patient’s therapeutically response, personalized medicine should be incorporated into current practice during the next years. 

It is necessary to understand better the molecular characteristics of ovarian cancer tumours and the new possible drugs or biological agents that could be useful in each patient. Based on this, immune checkpoints inhibitors, PARP inhibitors, and anti-angiogenic therapies constitute great advances for EOC treatment, and the testing of its combinations appears to be a reasonable alternative. Besides, there are several new possible and complementary therapies, including neurotrophic receptors inhibitors, repurposing drugs, non-coding RNAs, and the development of different nanoformulations under study, mainly in preclinical stages and pretend to offer new therapeutic alternatives for EOC patients. 

## Figures and Tables

**Figure 1 biomedicines-10-00077-f001:**
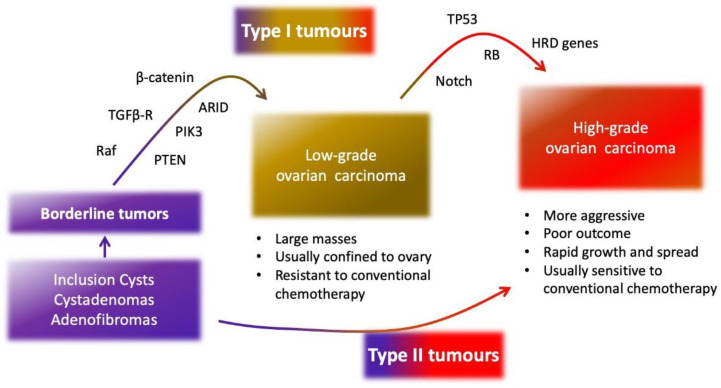
Main characteristics of type I and type II ovarian tumours. Type I tumours are characterized by sequential and low growth from the cyst and borderline tumours. In contrast, the evolution of type II tumours from pre-neoplastic lesions is quicker, resulting in an aggressive phenotype. Tumour evolution involves the acquisition of mutations in onco-suppressor genes as RAF kinase, beta-catenin, phosphatase and tensin homolog (PTEN), transforming growth factor-beta receptor (TGF-βR), phosphatidylinositol-4,5-bisphosphate 3-kinase (PIK3), AT-rich interactive domain-containing protein (ARID), TP53, retinoblastoma protein (RB), homologous recombinant deficiency (HRD) genes, and Notch pathway.

**Figure 2 biomedicines-10-00077-f002:**
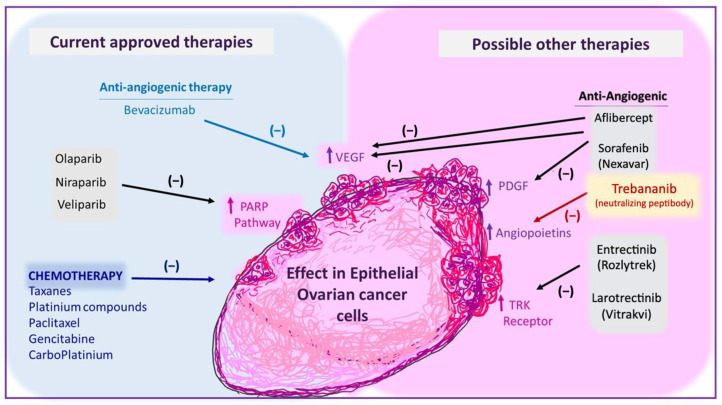
Summary of current and new possible therapies for ovarian cancer treatment. VEGF: vascular endothelial growth factor. PARP: poly adenosine diphosphate-ribose polymerases. TRK: tropomyosin receptor kinases. PDGF: platelet-derived growth factor.

**Figure 3 biomedicines-10-00077-f003:**
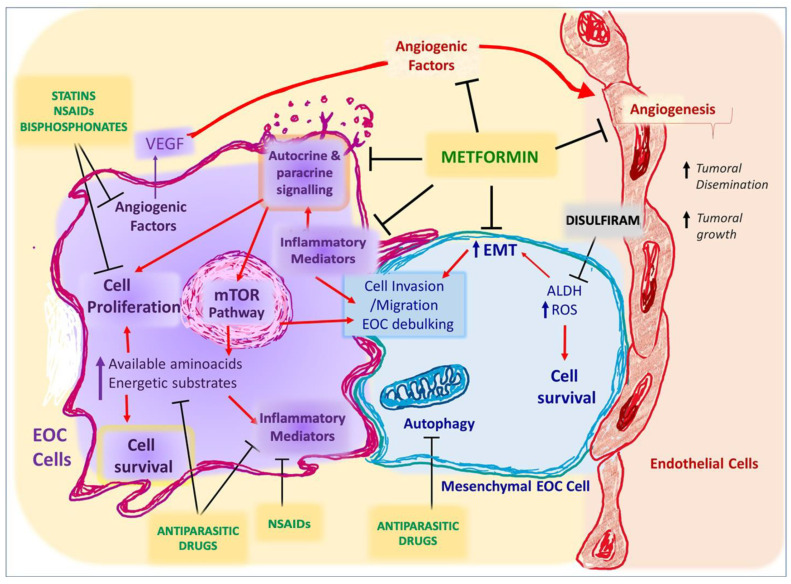
Repurposed drugs with anti-tumoral effects in epithelial ovarian cancer cells (in vitro and in vivo) and their molecular targets. EOC: epithelial ovarian cancer. VEGF: vascular endothelial growth factor. EMT: epithelial-mesenchymal transition. ALDH: aldehyde dehydrogenase. ROS: reactive oxygen species. NSAIDs: non-steroidal anti-inflammatories.

**Figure 4 biomedicines-10-00077-f004:**
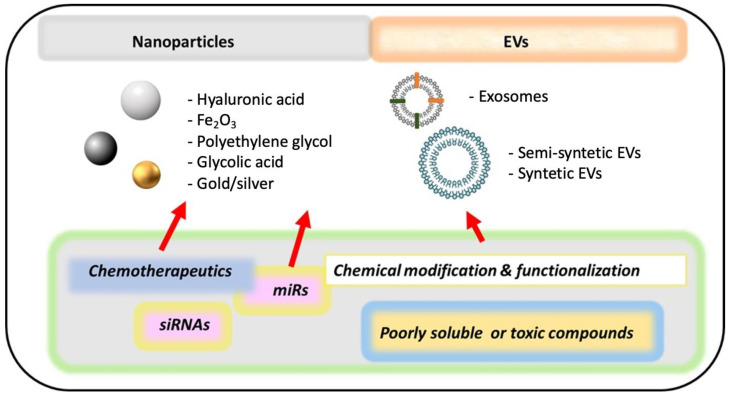
Some examples of advances in monotherapies and drug delivery for ovarian cancer. Most nanocarriers are under study in different models of ovarian cancer. It is expected that they could improve the delivery, half-life, and distribution of ovarian cancer therapies. Some examples of nanocarriers are nanoparticles, exosomes and modified extracellular vesicles (EVs). miRs: micro-RNAs. siRNAs: small interference RNAs.

**Table 2 biomedicines-10-00077-t002:** Summary of several new anti-angiogenic options under study in EOC.

Drugs	Mechanism	Study and Patients	Main Findings	Ref.
Trebananib (AMG 386)	Neutralizing peptibody that targets angiopoietin 1 and 2	Phase III study, tested with carboplatin and paclitaxel	Trebananib did not improve the progression-free survival of patients with advanced ovarian cancer	[106,107]
Sorafenib (Nexavar)	Protein kinase inhibitor of VEGF and PDGF receptors	Phase II study tested in combination with topotecan or bevacizumab	Clinical activity was observed in patients with ovarian cancer heavily-pretreated, bevacizumab-naive and platinum-resistant disease.	[108,109]
Entrectinib (Rozlytrek)	pan-TRK inhibitors (TRK receptors)	Phase I/II trials. At least one dose after standard treatments	Entrectinib was well tolerated and induced a durable response in patients with NTRK fusion-positive solid tumours.	[110]

**Table 3 biomedicines-10-00077-t003:** Summary of clinical trials using inhibitors of immune checkpoints with published results in ovarian cancer patients.

Drug	Study and Patients	Main Findings	Ref.
Niraparib in combination with pembrolizumab (anti-PD-1 antibody)	Phase I/II study in recurrent platinum-resistant ovarian cancer	The results of the combination were better than for single agents (ORR ^1^ was 18%). Antitumor activity was independent of BRCA mutation or HRD status and irrespective of PD-L1 expression	[78]
Pembrolizumab with cisplatin and gemcitabine	Phase II study in platinum-resistant ovarian cancer	Pembrolizumab addition did not appear to provide benefit beyond chemotherapy alone in the 18 patients treated.	[128]
SC-003 (anti-dipeptidase 3 antibody) and budigalimab (anti-DP-1)	Phase Ia/Ib in platinum-resistant/refractory ovarian cancer	Low and not durable responses in the 3 patients with the combined treatment. Low safety profile of SC-003	[129]
Pembrolizumab as single agent	Phase II study in patients with advanced and recurrent ovarian cancer	ORR of 7.4% in patients with one to three prior lines of treatment and 9.9% in patients with four or more lines of treatments. ORR 10.0% in patients with CPS ^2^ ≥ 10	[130,131]
Varlilumab (anti-CD27 antibody) and nivolumab (anti-PD-1 antibody)	Phase I/II study in patients advanced and refractory ovarian cancer	Increase in PD-L1 expression and CD8+ T cells in ovarian biopsies, changes related with a better outcome. Possible benefit in a group of resistant to PD-1 inhibitor monotherapy	[132,133]
Nivolumab and ipilimumab (anti-CTLA-4 antibody)	Phase II in patients with recurrent or persistent ovarian cancer	The combined use of nivolumab and ipilimumab in EOC showed a longer progression-free disease compared to nivolumab alone	[134]

^1^ ORR: overall response rate. ^2^ CPS: combined positive score.

**Table 4 biomedicines-10-00077-t004:** Summary of the main findings of studies using repurposing drugs for ovarian cancer treatment.

Drugs	Mechanism	Study and Patients	Main Findings	Ref.
Chloroquine	Autophagy inhibitor	Phase I/II study with advanced platinum-resistant epithelial ovarian cancer	Reverses cisplatin resistance in vitro. In patients, 30% expressed autophagy-related proteins but did not correlate with patient benefit	[152,153]
Ivermectin	Autophagy inhibitor	In vivo and in vitro studies	Synergistically suppresses tumour growth in combination with cisplatin or paclitaxel	[154,155]
Statins	HMG-CoA ^1^ reductase inhibitors	Observational studies	Statin use was inversely associated with ovarian cancer risk, particularly mucinous and endometroid subtypes	[156]
Bisphosphonates	Inhibitors of mevalonic acid pathway	In vitro studies	Zoledronate displayed additive and synergistic anti-tumoral effects with pitavastatin on cell growth, tumour-promoting cytokines, and mediators	[157,158]
Disulfiram	Aldehyde dehydrogenase inhibitor	Observational and in vitro studies	ALDH1A1 ^2^ -positive cells are negatively correlated with progression-free survival in HGS-EOC patients. In vitro enhancement of cisplatin-induced apoptosis	[159,160]
Arsenic trioxide	Pro-oxidative compound	In vitro and in vivo studies	Increases sensibility of ovarian cancer cells to PARP inhibitors and synergically suppress tumour growth with cisplatin and paclitaxel treatment	[161,162]
Metformin	mTOR ^3^ inhibitor	Observational studies in type 2 diabetic patients. Phase II study in non-diabetic patients	Decreases in ovarian cancer incidence and mortality in type 2 diabetic patients. Tumours from metformin-treated patients presented a decrease of cancer stem cells markers and an increased sensitivity to cisplatin ex vivo.	[163,164,165]
NSAIDs ^4^	COX ^5^ inhibitors	In vitro and in vivo studies	Anti-inflammatory effects. Increases paclitaxel sensitivity and restores cisplatin sensitivity	[166,167,168]

^1^ HMG-CoA: β-Hydroxy β-methylglutaryl-coenzyme A. ^2^ ALDH1A1: aldehyde dehydrogenase 1 family member A1. ^3^ mTOR: mammalian target of rapamycin. ^4^ NSAIDs: non-steroidal anti-inflammatory drugs. ^5^ COX: cyclooxygenase.

## Data Availability

Not applicable.

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
