# Peer review of "Current Treatments and New Possible Complementary Therapies for Epithelial Ovarian Cancer"

_biomedicines, 2021, doi:10.3390/biomedicines10010077_

Round 1

Reviewer 1 Report

The review article by Garrido et al entitled “Current treatments and new possible complementary therapies 2 for epithelial ovarian cancer” provides a broad overview of ovarian cancer therapy. This includes standard chemotherapy, PARP inhibitors, inhibitors of angiogenesis and multiple other candidate drugs in various stages of preclinical and clinical study. Given the decision to emphasize breadth rather than depth in this review, there are some surprising omissions such as the immune checkpoint inhibitors that should be addressed.

Due to the scope of the article there is little new information on PARP inhibitors and inhibitors of angiogenesis that adds to the abundant review literature published in 2021 on these ovarian cancer therapeutics. Furthermore, the authors state several times that introduction of PARP inhibitors and anti-angiogenic therapies “did not have a significant impact on patients’ survival”.  These statements are contradicted by language in the text and some reported results from clinical studies. This controversial statement is not adequately supported by the authors.  A more thorough and thoughtful treatment of the benefits (or lack thereof) of PARP inhibitors and anti-angiogenic agents is needed.

The review offers relatively new information in the sections on repurposed drugs, non-coding RNAs and nanomedicine but these topics are covered rather superficially. For example the authors conclude the section with “the use of statins as complementary therapy for EOC could not be discarded, especially if it is possible to select those patients who may have a more significant benefit from their use” without addressing what target or targets might be considered for patient selection. Furthermore, there is recent literature missing from the section including a relevant meta-analysis (doi: 10.1007/s10552-020-01327-8 as an example).

Overall, the review content is not in good alignment with the concluding paragraph (Main Conclusions) that emphasizes new potential drugs or biological agents.

Specific comments:

There are many instances of confusing or awkward language that interfere with reader comprehension. This is an issue throughout the document. A few examples include:

Paragraph lines 71-82 is confusing. There is a lack of clarity regarding ovarian surface epithelium vs fallopian tube origins of types of ovarian cancer.

Line 94 These findings give a hint that molecular characteristics of each patient with EOC should be more thoughtful to choose the best existing treatment

Line 169-171 Gemcitabine is a synthetic nucleoside inhibitor which incremented the platinum-DNA adducts incorporation and inhibits the repair of cisplatin intra-strand adducts[61-63], whose proposal is to enhance EOC response to cisplatin treatment.

Additional summary tables would benefit the reader.

A table of FDA approved drugs for angiogenesis and PARP inhibitor with generic name, trade name(s), date of US FDA approval and approved use.

A table comparable to the PARP inhibitor table (Table 1) to cover clinical studies should be developed for the angiogenesis inhibitors.

Current Table 2 with new inhibitors is not comparable to Table 1 for information provided.

There should be a table with the clinical trials related to repurposed drugs to supplement the text.

Author Response

First, we would like to thank the reviewer for the revision and comments

Point-by-point response to reviewers´ comments:

1.1. The review article by Garrido et al entitled “Current treatments and new possible complementary therapies for epithelial ovarian cancer” provides a broad overview of ovarian cancer therapy. This includes standard chemotherapy, PARP inhibitors, inhibitors of angiogenesis and multiple other candidate drugs in various stages of preclinical and clinical study. Given the decision to emphasize breadth rather than depth in this review, there are some surprising omissions such as the immune checkpoint inhibitors that should be addressed.

Answer:

To improve the manuscript, we included a section with the main advances of immune checkpoints in the context of ovarian cancer, which is highlighted in yellow. Please, see the new section 7 “Immune checkpoint inhibitors as an alternative for ovarian cancer treatment”.

1.2. Due to the scope of the article there is little new information on PARP inhibitors and inhibitors of angiogenesis that adds to the abundant review literature published in 2021 on these ovarian cancer therapeutics. Furthermore, the authors state several times that introduction of PARP inhibitors and anti-angiogenic therapies “did not have a significant impact on patients’ survival”.  These statements are contradicted by language in the text and some reported results from clinical studies. This controversial statement is not adequately supported by the authors.  A more thorough and thoughtful treatment of the benefits (or lack thereof) of PARP inhibitors and anti-angiogenic agents is needed.

Answer:

We agree with this comment. In order to improve these points, we performed a second revision about the state of the art of PARP inhibitors and angiogenic inhibitors in ovarian cancer, hence, new information was incorporated. Please, check the highlighted paragraphs in the sections 4.2 and 4.3 of the manuscript.

1.3. The review offers relatively new information in the sections on repurposed drugs, non-coding RNAs and nanomedicine but these topics are covered rather superficially. For example the authors conclude the section with “the use of statins as complementary therapy for EOC could not be discarded, especially if it is possible to select those patients who may have a more significant benefit from their use” without addressing what target or targets might be considered for patient selection. Furthermore, there is recent literature missing from the section including a relevant meta-analysis (doi: 10.1007/s10552-020-01327-8 as an example).

Answer:

We thank the reviewer for this information, which is relevant for the current work. The analysis required in the section of statins was added to the manuscript. Please, check the highlighted paragraph in the section 8.2.

Regarding the shallowness with which we dressed the topics of biological compounds and drug repurposing, we want to emphasize that the focus of the work is to give a general overview of the advances in existing therapies and new possible therapies for ovarian cancer. Each of these topics is quite extensive and a separate review could be carried out. Therefore, our purpose was to give only an initial sight about these topics.

1.4. Overall, the review content is not in good alignment with the concluding paragraph (Main Conclusions) that emphasizes new potential drugs or biological agents.

Answer:

We reviewed and modified the conclusion section. Please, check the changes, which are highlighted in yellow.

1.5. There are many instances of confusing or awkward language that interfere with reader comprehension. This is an issue throughout the document. A few examples include:

Paragraph lines 71-82 is confusing. There is a lack of clarity regarding ovarian surface epithelium vs fallopian tube origins of types of ovarian cancer.

Line 94 These findings give a hint that molecular characteristics of each patient with EOC should be more thoughtful to choose the best existing treatment

Line 169-171 Gemcitabine is a synthetic nucleoside inhibitor which incremented the platinum-DNA adducts incorporation and inhibits the repair of cisplatin intra-strand adducts[61-63], whose proposal is to enhance EOC response to cisplatin treatment.

Answer:

We checked these sentences and performed the required improvements regarding language. In addition, we carried out a new revision of the manuscript to correct any possible mistakes and confusing sentences.

1.6. Additional summary tables would benefit the reader.

A table of FDA approved drugs for angiogenesis and PARP inhibitor with generic name, trade name(s), date of US FDA approval and approved use.

A table comparable to the PARP inhibitor table (Table 1) to cover clinical studies should be developed for the angiogenesis inhibitors.

Current Table 2 with new inhibitors is not comparable to Table 1 for information provided.

There should be a table with the clinical trials related to repurposed drugs to supplement the text.

Answer:

We modified the original tables and added new tables according to the reviewer´s requirements.  Please, check the tables highlighted in yellow along the manuscript.

Reviewer 2 Report

Dear Authors,

My comments:

  1. "anti-tumoral" or "anti-tumour", please check what will better
  2. Table describing EOC type I and II may be useful.
  3. Please, check references..some not full references 

Author Response

First, we would like to thank the reviewer for the revision and comments.

Point-by-point response to reviewers´ comments

1. "anti-tumoral" or "anti-tumour", please check what will better

Answer:

We would like to thank the reviewer for your revision and comments. The manuscript was reviewed to include a unique term (anti-tumoral).

2. Table describing EOC type I and II may be useful.

Answer:

To better understand the main characteristics of these tumours, we included a new figure. Please, see the new figure 1 in the section 3.1 of the manuscript.

3. Please, check references. Some not full references 

Answer:

We performed a new revision of the reference list and mistakes were corrected. In addition, please new references were included, particularly from studies published during the last 2 years.

Round 2

Reviewer 1 Report

The revised article by Garrido et al is responsive to many of the critiques and requests from the previous review including the addition of a section on immune checkpoint inhibitors and summary tables.

The text continues to have contradictory statements regarding the impact of therapeutics (in particular PARP and angiogenesis inhibitors) on ovarian cancer patient survival. The abstract states that these therapies did not have an impact on patient survival, but other statements, including paragraph 1, state the opposite. I believe greater clarity could be achieved by using specific language for overall survival versus progression free survival whenever patient survival is noted in the context of a therapeutic.

There are many instances of language usage, spelling or confusing language in the document. The following instances are not comprehensive as there are others that are not caught in this list.

Lines: 45, 75,95-96, 102, 191 (should specify country of FDA), 218-223, 252,263,318,433,447,464 (colon shouldn't end a paragraph), 483,495 (drug capitalization not consistent), 509-510 (appears to be a misstatement that pravastatin use increases ovarian cancer risk),522-526,538,550-553, 570 (unclear what "their" is referencing",578-579, 593 (why drug in bold?), 677-678, 701,710, 769,772, 773-774 (statement suggests iron nanoparticles are used to treat patients), 789, 797-798, 829,838.

Author Response

Point-by-point response to reviewer’s comments

REVIEWER 1

  1. The text continues to have contradictory statements regarding the impact of therapeutics (in particular PARP and angiogenesis inhibitors) on ovarian cancer patient survival. The abstract states that these therapies did not have an impact on patient survival, but other statements, including paragraph 1, state the opposite. I believe greater clarity could be achieved by using specific language for overall survival versus progression free survival whenever patient survival is noted in the context of a therapeutic.

 Answer:

We would like to thank again the reviewer for the revision and comments. As recommended, we used the specific terms related to survival in the new version of the manuscript and changed this sentence in the abstract. Please check the changes highlighted in yellow.

  1. There are many instances of language usage, spelling or confusing language in the document. The following instances are not comprehensive as there are others that are not caught in this list.

Lines: 45, 75,95-96, 102, 191 (should specify country of FDA), 218-223, 252,263,318,433,447,464 (colon shouldn't end a paragraph),

483,495 (drug capitalization not consistent),

Answer:

We reviewed the manuscript to detect and correct inconsistencies in language.

  1. 509-510 (appears to be a misstatement that pravastatin use increases ovarian cancer risk),

Answer:

The sentence was removed from the manuscript in order to avoid confusion.

  1. 522-526,538,550-553, 570 (unclear what "their" is referencing",578-579, 593 (why drug in bold?)

Answer:

The format was revised and corrected. Bibliographies were added to the manuscript.

  1. 677-678, 701,710, 769,772, 773-774 (statement suggests iron nanoparticles are used to treat patients), 789, 797-798, 829,838.

Answer:

The respective clarifications were added in section 10 “New methods of drug delivery (nanomedicine)”. Please, check the changes highlighted in yellow.